# QUANTIFYING STATISTICAL SIGNIFICANCE OF NEURAL NETWORK REPRESENTATION-DRIVEN HYPOTHESES BY SELECTIVE INFERENCE

## ABSTRACT

In the past few years, various approaches have been developed to explain and interpret deep neural network (DNN) representations, but it has been pointed out that these representations are sometimes unstable and not reproducible. In this paper, we interpret these representations as hypotheses driven by DNN (called DNN-driven hypotheses) and propose a method to quantify the reliability of these hypotheses in statistical hypothesis testing framework. To this end, we introduce Selective Inference (SI) framework, which has received much attention in the past few years as a new statistical inference framework for data-driven hypotheses. The basic idea of SI is to make conditional inferences on the selected hypotheses under the condition that they are selected. In order to use SI framework for DNN representations, we develop a new SI algorithm based on homotopy method which enables us to derive the exact (non-asymptotic) conditional sampling distribution of the DNN-driven hypotheses. In this paper, we demonstrate the proposed method in computer vision tasks as practical examples. We conduct experiments on both synthetic and real-world datasets, through which we offer evidence that our proposed method can successfully control the false positive rate, has decent performance in terms of computational efficiency, and provides good results in practical applications.

## 1 INTRODUCTION

The remarkable predictive performance of deep neural networks (DNNs) stems from their ability to learn appropriate representations from data. In order to understand the decision-making process of DNNs, it is thus important to be able to explain and interpret DNN representations. For example, in image classification tasks, knowing the *attention region* from DNN representation allows us to understand the reason for the classification. In the past few years, several methods have been developed to explain and interpret DNN representations (Ribeiro et al., 2016; Bach et al., 2015; Doshi-Velez & Kim, 2017; Lundberg & Lee, 2017; Zhou et al., 2016; Selvaraju et al., 2017); however, some of them have turned out to be unstable and not reproducible (Kindermans et al., 2017; Ghorbani et al., 2019; Melis & Jaakkola, 2018; Zhang et al., 2020; Dombrowski et al., 2019; Heo et al., 2019). Therefore, it is crucially important to develop a method to quantify the reliability of DNN representations.

In this paper, we interpret these representations as hypotheses that are driven by DNN (called DNN-driven hypotheses) and employ statistical hypothesis testing framework to quantify the reliability of DNN representations. For example, in an image classification task, the reliability of an attention region can be quantified based on the statistical significance of the difference between the attention region and the rest of the image. Unfortunately, however, traditional statistical test cannot be applied to this problem because the hypothesis (attention region in the above example) itself is selected by the data. Traditional statistical test is valid only when the hypothesis is non-random. Roughly speaking, if a hypothesis is selected by the data, the hypothesis will over-fit to the data and the bias needs to be corrected when assessing the reliability of the hypothesis.

Our main contribution in this paper is to introduce *Selective Inference (SI)* approach for testing the reliability of DNN representations. The basic idea of SI is to perform statistical inference under the condition that the hypothesis is selected. SI approach has been demonstrated to be effective

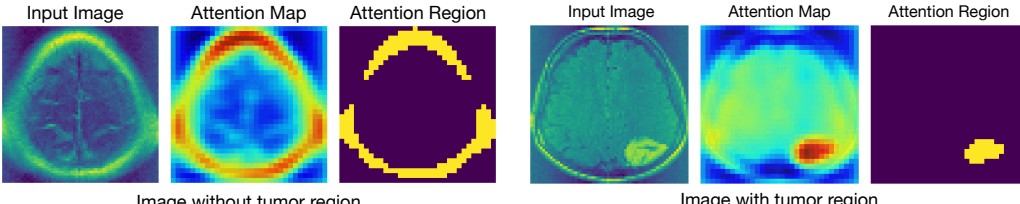

(a) naive-$p$ = **0.00** (false positive—wrong detection) and selective-$p$ = **0.94** (true negative)

(b) naive-$p$ = **0.00** (true positive) and selective-$p$ = **0.02** (true positive)

Figure 1: Examples of the proposed method on brain tumor image classification. Given a CNN trained to classify tumor versus non-tumor brain images in advance, our method provides the statistical significance of the attention region for each test image in the form of $p$-values by comparing the pixel information in the attention and non-attention regions. Since the attention region is selected by the input image, the $p$-value obtained by the naive comparison of the two regions (naive $p$-value) is highly biased. In the left-hand side figure where there is no brain tumor, the naive $p$-value is nearly zero (indicating false positive—incorrectly identifying tumor region), while the proposed selective $p$-value is large (indicating true negative). On the other hand, in the right-hand side figure where there actually exist a brain tumor, both the naive $p$-value and the selective $p$-values are very small (indicating true positive). The proposed selective inference method can provide valid exact (non-asymptotic) $p$-values for DNN representations such as attentions.

in the context of feature selections such as Lasso. In this paper, in order to introduce SI for DNN representations, we develop a novel SI algorithm based on homotopy method, which enables us to derive the exact (non-asymptotic) conditional sampling distribution of the DNN-driven hypothesis. We use $p$-value as a criterion to quantify the reliability of DNN representation. In the literature, $p$-values are often misinterpreted and there are various source of mis-interpretation has been discussed (Wasserstein & Lazar, 2016). In this paper, by using SI, we address one of the sources of mis-interpreted $p$-values; the p-values are biased when the hypothesis is selected after looking at the data (often called double-dipping or data dredging). We believe our approach is a first significant step to provide *valid* $p$-values for assessing the reliability of DNN representations. Figure 1 shows an example that illustrates the importance of our method.

**Related works.** Several recent approaches have been developed to visualize and understand a trained DNN. Many of these post-hoc approaches (Mahendran & Vedaldi, 2015; Zeiler & Fergus, 2014; Dosovitskiy & Brox, 2016; Simonyan et al., 2013) have focused on developing visualization tools for the activation maps and/or the filter weights within trained networks. Others have aimed to identify the discriminative regions in an input image, given a trained network (Selvaraju et al., 2017; Fong & Vedaldi, 2017; Zhou et al., 2016; Lundberg & Lee, 2017). In parallel, some recent studies have showed that many popular methods for explanation and interpretation are not stable with respect to the perturbation or the adversarial attack on the input data and the model (Kindermans et al., 2017; Ghorbani et al., 2019; Melis & Jaakkola, 2018; Zhang et al., 2020; Dombrowski et al., 2019; Heo et al., 2019). However, there are no previous studies that quantitatively evaluate the stability and reproducibility of DNN representations with a rigorous statistical inference framework.

In the past few years, SI has been actively studied for inference on the features of linear models selected by several feature selection methods, e.g., Lasso (Lee et al., 2016; Liu et al., 2018; Duy & Takeuchi, 2020). The basic idea of SI is to make inference conditional on the selection event, which allows us to derive the exact (non-asymptotic) sampling distribution of the test statistic. Besides, SI has also been applied to various problems (Bachoc et al., 2014; Fithian et al., 2015; Choi et al., 2017; Tian et al., 2018; Chen & Bien, 2019; Hyun et al., 2018; Bachoc et al., 2018; Loftus & Taylor, 2014; Loftus, 2015; Panigrahi et al., 2016; Tibshirani et al., 2016; Yang et al., 2016; Suzumura et al., 2017; Duy et al., 2020). However, to the best of our knowledge, there is no existing study that provides SI for DNNs, which is technically challenging. This study is partly motivated by Tanizaki et al. (2020) where the authors provide a framework to compute $p$-values for image segmentation results provided by graph cut and threshold-based segmentation algorithms. As we demonstrate in this paper, our method can be also used to assess the reliability of DNN-based segmentation results.

**Contribution.** To our knowledge, this is the first study that provides an exact (non-asymptotic) inference method for statistically quantifying the reliability of data-driven hypotheses that are discovered from DNN representation. We propose a novel SI *homotopy method*, inspired by Duy & Takeuchi (2020), for conducting powerful and efficient SI for DNN representations. We conduct experiments on both synthetic and real-world datasets, through which we offer evidence that our proposed method can successfully control the false positive rate, has decent performance in terms of computational efficiency, and provides good results in practical applications. We provide our implementation in the supplementary document and it will be released when this paper is published.

## 2 PROBLEM STATEMENT

To formulate the problem, we denote an image with $n$ pixels corrupted with Gaussian noise as

$$\boldsymbol{X} = (X_1, ..., X_n)^\top = \boldsymbol{\mu} + \boldsymbol{\varepsilon}, \quad \boldsymbol{\varepsilon} \sim \mathbb{N}(\boldsymbol{0}, \Sigma), \tag{1}$$

where $\boldsymbol{\mu} \in \mathbb{R}^n$ is an unknown mean pixel intensity vector and $\boldsymbol{\varepsilon} \in \mathbb{R}^n$ is a vector of Normally distributed noise with the covariance matrix $\Sigma$ that is known or able to be estimated from external data. We note that we do not assume that the pixel intensities in an image follow Normal distribution in Equation (1). Instead, we only assume that the vector of noises added to the true pixel values follows a multivariate Normal distribution. For an image $\boldsymbol{X}$ and a trained DNN, the main target is to identify an *attention region* (discriminative/informative region) in the input image $\boldsymbol{X}$ based on a DNN representation. A pixel is assigned to the attention region if its corresponding value in the representation layer is greater than a pre-defined threshold. We denote the set of pixels of $\boldsymbol{X}$ divided into attention region and non-attention region as $\mathcal{C}_{\boldsymbol{X}}^+$ and $\mathcal{C}_{\boldsymbol{X}}^-$, respectively.

**Definition 1.** *We define $\mathcal{A}(\boldsymbol{X})$ as the event that the result of dividing pixels of image $\boldsymbol{X}$ into two sets of pixels $\mathcal{C}_{\boldsymbol{X}}^+$ and $\mathcal{C}_{\boldsymbol{X}}^-$ is obtained by applying a DNN on $\boldsymbol{X}$, i.e.,*

$$\mathcal{A}(\boldsymbol{X}) = \{\mathcal{C}_{\boldsymbol{X}}^+, \mathcal{C}_{\boldsymbol{X}}^-\}. \tag{2}$$

**Quantifying the statistical significance of DNN-driven hypotheses.** Given an observed image $\boldsymbol{x}^{\mathrm{obs}} \in \mathbb{R}^n$ sampled from the model (1), we can obtain $\mathcal{C}_{\boldsymbol{x}^{\mathrm{obs}}}^+$ and $\mathcal{C}_{\boldsymbol{x}^{\mathrm{obs}}}^-$ by applying DNN on $\boldsymbol{x}^{\mathrm{obs}}$. Let us consider a score $\Delta$ that represents the degree to which the attention region differs from the non-attention region. In general, we can define any score as long as it is written in the form $\Delta = \boldsymbol{\eta}^\top \boldsymbol{x}^{\mathrm{obs}}$. For example, we can define $\Delta$ as the difference in average pixel values between the attention region and the non-attention region, i.e.,

$$\Delta = m_{\mathcal{C}_{\boldsymbol{x}^{\mathrm{obs}}}^+} - m_{\mathcal{C}_{\boldsymbol{x}^{\mathrm{obs}}}^-} = \frac{1}{|\mathcal{C}_{\boldsymbol{x}^{\mathrm{obs}}}^+|} \sum_{i \in \mathcal{C}_{\boldsymbol{x}^{\mathrm{obs}}}^+} x_i^{\mathrm{obs}} - \frac{1}{|\mathcal{C}_{\boldsymbol{x}^{\mathrm{obs}}}^-|} \sum_{i \in \mathcal{C}_{\boldsymbol{x}^{\mathrm{obs}}}^-} x_i^{\mathrm{obs}} = \boldsymbol{\eta}^\top \boldsymbol{x}^{\mathrm{obs}},$$

where $\boldsymbol{\eta} = \frac{1}{|\mathcal{C}_{\boldsymbol{x}^{\mathrm{obs}}}^+|} \mathbf{1}_{\mathcal{C}_{\boldsymbol{x}^{\mathrm{obs}}}^+}^n - \frac{1}{|\mathcal{C}_{\boldsymbol{x}^{\mathrm{obs}}}^-|} \mathbf{1}_{\mathcal{C}_{\boldsymbol{x}^{\mathrm{obs}}}^-}^n$, and $\mathbf{1}_{\mathcal{C}}^n \in \mathbb{R}^n$ is a vector whose elements belonging to a set $\mathcal{C}$ are 1, and 0 otherwise.

If the value of $|\Delta|$ is sufficiently large, the difference between $\mathcal{C}_{\boldsymbol{x}^{\mathrm{obs}}}^+$ and $\mathcal{C}_{\boldsymbol{x}^{\mathrm{obs}}}^-$ is significant and the attention region is reliable. To quantify the statistical significance, we consider a statistical hypothesis testing with the following null hypothesis $\mathrm{H}_0$ and alternative hypothesis $\mathrm{H}_1$:

$$\mathrm{H}_0 : \mu_{\mathcal{C}_{\boldsymbol{x}^{\mathrm{obs}}}^+} = \mu_{\mathcal{C}_{\boldsymbol{x}^{\mathrm{obs}}}^-} \quad \text{vs.} \quad \mathrm{H}_1 : \mu_{\mathcal{C}_{\boldsymbol{x}^{\mathrm{obs}}}^+} \neq \mu_{\mathcal{C}_{\boldsymbol{x}^{\mathrm{obs}}}^-}, \tag{3}$$

where $\mu_{\mathcal{C}_{\boldsymbol{x}^{\mathrm{obs}}}^+}$ and $\mu_{\mathcal{C}_{\boldsymbol{x}^{\mathrm{obs}}}^-}$ are the true means of the pixel values in the attention region and non-attention region, respectively. Given a significance level $\alpha$ (e.g., 0.05), we reject $\mathrm{H}_0$ if the $p$-value is smaller than $\alpha$, which indicates the attention region differs from the non-attention region. Otherwise, we cannot say that the difference is significant.

In a standard (naive) statistical test, the hypotheses in (3) are assumed to be fixed, i.e., non-random. Then, the naive (two-sided) $p$-value is simply given as

$$p_{\mathrm{naive}} = \mathbb{P}_{\mathrm{H}_0} \left( |\boldsymbol{\eta}^\top \boldsymbol{X}| \geq |\Delta| \right) = \mathbb{P}_{\mathrm{H}_0} \left( |\boldsymbol{\eta}^\top \boldsymbol{X}| \geq |\boldsymbol{\eta}^\top \boldsymbol{x}^{\mathrm{obs}}| \right). \tag{4}$$

However, since the hypotheses in (3) are actually not fixed in advance, the naive $p$-value is *not valid* in the sense that, if we reject $\mathrm{H}_0$ with a significance level $\alpha$, the false detection rate (type-I error) cannot be controlled at level $\alpha$, which indicates that $p_{\mathrm{naive}}$ is unreliable. This is due to the fact that the hypotheses (the attention region) in (3) are *selected* by looking at the data (the input image), and thus *selection bias* exists. This selection bias is sometimes called *data dredging*, *data snooping* or *p-hacking* (Ioannidis, 2005; Head et al., 2015).

**Selective inference (SI) for computing valid $p$-values.** The basic idea of SI is to make inference *conditional* on the selection event, which allows us to derive the *exact* (non-asymptotic) sampling distribution of the test statistic $\boldsymbol{\eta}^\top \boldsymbol{X}$ in an attempt to avoid the selection bias. Thus, we employ the following *conditional* $p$-value

$$p_{\text{selective}} = \mathbb{P}_{\text{H}_0}\left(|\boldsymbol{\eta}^\top \boldsymbol{X}| \geq |\boldsymbol{\eta}^\top \boldsymbol{x}^{\text{obs}}| \mid \mathcal{A}(\boldsymbol{X}) = \mathcal{A}(\boldsymbol{x}^{\text{obs}}), \boldsymbol{q}(\boldsymbol{X}) = \boldsymbol{q}(\boldsymbol{x}^{\text{obs}})\right), \qquad (5)$$

where $\boldsymbol{q}(\boldsymbol{X}) = (I_n - \boldsymbol{c}\boldsymbol{\eta}^\top)\boldsymbol{X}$ with $\boldsymbol{c} = \Sigma\boldsymbol{\eta}(\boldsymbol{\eta}^\top \Sigma \boldsymbol{\eta})^{-1}$. The first condition $\mathcal{A}(\boldsymbol{X}) = \mathcal{A}(\boldsymbol{x}^{\text{obs}})$ indicates the event that the result of dividing pixels into an attention region and non-attention region for a random image $\boldsymbol{X}$ is the same as that of the observed image $\boldsymbol{x}^{\text{obs}}$, i.e., $\mathcal{C}_{\boldsymbol{X}}^+ = \mathcal{C}_{\boldsymbol{x}^{\text{obs}}}^+$ and $\mathcal{C}_{\boldsymbol{X}}^- = \mathcal{C}_{\boldsymbol{x}^{\text{obs}}}^-$. The second condition $\boldsymbol{q}(\boldsymbol{X}) = \boldsymbol{q}(\boldsymbol{x}^{\text{obs}})$ indicates the component that is independent of the test statistic for $\boldsymbol{X}$ is the same as the one for $\boldsymbol{x}^{\text{obs}}$. The $\boldsymbol{q}(\boldsymbol{X})$ corresponds to the component $\boldsymbol{z}$ in the seminal SI paper of Lee et al. (2016) (Sec 5, Eq 5.2 and Theorem 5.2). The $p$-value in (5), which is called *selective type I error* or *selective p-values* in the SI literature (Fithian et al., 2014), is *valid* in the sense that $\mathbb{P}_{\text{H}_0}(p_{\text{selective}} < \alpha) = \alpha, \forall \alpha \in [0, 1]$, i.e., the false detection rate is theoretically controlled at level $\alpha$ indicating the selective $p$-value is reliable.

To calculate the selective $p$-value in (5), we need to identify the conditional data space. Let us define the set of $\boldsymbol{x} \in \mathbb{R}^n$ that satisfies the conditions in (5) as

$$\mathcal{X} = \{\boldsymbol{x} \in \mathbb{R}^n \mid \mathcal{A}(\boldsymbol{x}) = \mathcal{A}(\boldsymbol{x}^{\text{obs}}), \boldsymbol{q}(\boldsymbol{x}) = \boldsymbol{q}(\boldsymbol{x}^{\text{obs}})\}. \qquad (6)$$

According to the second condition, the data in $\mathcal{X}$ are restricted to a line (Sec 6 in Liu et al. (2018), and Fithian et al. (2014)). Therefore, the set $\mathcal{X}$ can be re-written, using a scalar parameter $z \in \mathbb{R}$, as

$$\mathcal{X} = \{\boldsymbol{x}(z) = \boldsymbol{a} + \boldsymbol{b}z \mid z \in \mathcal{Z}\}, \qquad (7)$$

where $\boldsymbol{a} = \boldsymbol{q}(\boldsymbol{x}^{\text{obs}})$, $\boldsymbol{b} = \Sigma\boldsymbol{\eta}_\ell(\boldsymbol{\eta}_\ell^\top \Sigma \boldsymbol{\eta}_\ell)^{-1}$, and

$$\mathcal{Z} = \left\{z \in \mathbb{R} \mid \mathcal{A}(\boldsymbol{x}(z)) = \mathcal{A}(\boldsymbol{x}^{\text{obs}})\right\}. \qquad (8)$$

Now, let us consider a random variable $Z \in \mathbb{R}$ and its observation $z^{\text{obs}} \in \mathbb{R}$ that satisfy $\boldsymbol{X} = \boldsymbol{a} + \boldsymbol{b}Z$ and $\boldsymbol{x}^{\text{obs}} = \boldsymbol{a} + \boldsymbol{b}z^{\text{obs}}$. Then, the selective $p$-value in (5) is re-written as

$$p_{\text{selective}} = \mathbb{P}_{\text{H}_0}\left(|\boldsymbol{\eta}^\top \boldsymbol{X}| \geq |\boldsymbol{\eta}^\top \boldsymbol{x}^{\text{obs}}| \mid \boldsymbol{X} \in \mathcal{X}\right) = \mathbb{P}_{\text{H}_0}\left(|Z| \geq |z^{\text{obs}}| \mid Z \in \mathcal{Z}\right). \qquad (9)$$

Since the variable $Z \sim \mathbb{N}(0, \boldsymbol{\eta}^\top \Sigma \boldsymbol{\eta})$ under the null hypothesis, the law of $Z \mid Z \in \mathcal{Z}$ follows a truncated Normal distribution. Once the truncation region $\mathcal{Z}$ is identified, the selective $p$-value (9) can be computed as

$$p_{\text{selective}} = F_{0,\boldsymbol{\eta}^\top \Sigma \boldsymbol{\eta}}^{\mathcal{Z}}(-|z^{\text{obs}}|) + 1 - F_{0,\boldsymbol{\eta}^\top \Sigma \boldsymbol{\eta}}^{\mathcal{Z}}(|z^{\text{obs}}|), \qquad (10)$$

where $F_{m,s^2}^{\mathcal{E}}$ is the c.d.f. of the truncated normal distribution with mean $m$, variance $s^2$ and truncation region $\mathcal{E}$. Therefore, the most important task is to identify $\mathcal{Z}$.

**Extension of the problem setup to hypothesis driven from DNN-based image segmentation.** We interpret the hypothesis driven from image segmentation result as the one obtained from the *representation at output layer* instead of internal representation. Our problem setup is general and can be directly applied to this case. For example, we can consider the attention region as the object region and the non-attention region as the background region. Then, we can conduct SI to quantify the significance of the difference between object and background regions. We note that we consider the case where the image is segmented into two regions—object and background—to simplify the problem and notations. The extension to more than two regions is straightforward.

## 3 PROPOSED METHOD

As we discussed in §2, to calculate the selective $p$-value, the truncation region $\mathcal{Z}$ in Equation (8) must be identified. To construct $\mathcal{Z}$, we have to 1) compute $\mathcal{A}(\boldsymbol{x}(z))$ for all $z \in \mathbb{R}$, and 2) identify the set of intervals of $z$ on which $\mathcal{A}(\boldsymbol{x}(z)) = \mathcal{A}(\boldsymbol{x}^{\text{obs}})$. However, it seems intractable to obtain $\mathcal{A}(\boldsymbol{x}(z))$ for infinitely many values of $z \in \mathbb{R}$.

Our first idea to develop SI for DNN is that we additionally condition on some extra event to make the problem tractable. We now focus on a class of DNNs whose activation functions (AFs) are piecewise-linear, e.g., ReLU, Leaky ReLU (the extension to general AFs is discussed later). Then, we consider additionally conditioning on the selected piece of each piecewise-linear AF in the DNN.

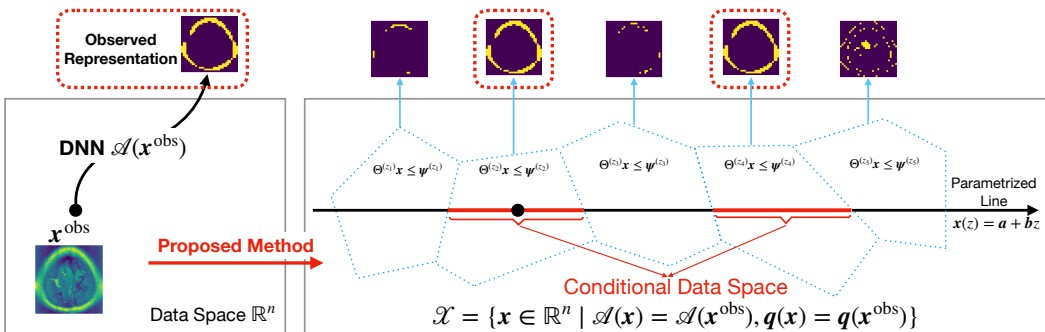

Figure 2: A schematic illustration of the proposed method. By applying DNN on the observed image $\boldsymbol{x}^{\mathrm{obs}}$, we obtain an representation. Then, we parametrize $\boldsymbol{x}^{\mathrm{obs}}$ with a scalar parameter $z$ in the dimension of test-statistic to identify the subspace $\mathcal{X}$ whose data has the *same representation* as $\boldsymbol{x}^{\mathrm{obs}}$ has. Finally, the valid statistical inference is conducted conditional on $\mathcal{X}$. We introduce a homotopy method for efficiently characterizing the conditional data space $\mathcal{X}$.

**Definition 2.** *Let $s_j(\boldsymbol{x})$ be "the selected piece" of a piecewise-linear AF at the $j$-th unit in a DNN for a given input image $\boldsymbol{x}$, and let $\boldsymbol{s}(\boldsymbol{x})$ be the set of $s_j(\boldsymbol{x})$ for all the nodes in a DNN .*

For example, for a ReLU activation function, $s_j(\boldsymbol{x})$ takes either 0 or 1 depending on whether the input to the $j$-th unit is located at the flat part (inactive) or the linear part (active) of the ReLU function. Using the notion of selected pieces $\boldsymbol{s}(x)$, instead of computing the selective $p$-value in (9), we consider the following *over-conditioning (oc)* conditional $p$-value

$$p_{\mathrm{selective}}^{\mathrm{oc}} = \mathbb{P}_{\mathrm{H}_0}\left(|Z| \geq |z^{\mathrm{obs}}| \mid Z \in \mathcal{Z}^{\mathrm{oc}}\right), \tag{11}$$

where $\mathcal{Z}^{\mathrm{oc}} = \left\{z \in \mathbb{R} \mid \mathcal{A}(\boldsymbol{x}(z)) = \mathcal{A}(\boldsymbol{x}^{\mathrm{obs}}), \boldsymbol{s}(\boldsymbol{x}(z)) = \boldsymbol{s}(\boldsymbol{x}^{\mathrm{obs}})\right\}$. However, such an over-conditioning in SI leads to the loss of statistical power (Lee et al., 2016).

Our second idea is to develop a *homotopy method* to resolve the over-conditioning problem, i.e., remove the conditioning of $\boldsymbol{s}(\boldsymbol{x}(z)) = \boldsymbol{s}(\boldsymbol{x}^{\mathrm{obs}})$. With the homotopy method, we can efficiently compute $\mathcal{A}(\boldsymbol{x}(z))$ in a finite number of operations without the need of considering infinitely many values of $z \in \mathbb{R}$, which is subsequently used to obtain truncation region $\mathcal{Z}$ in (8). The main idea is to compute a finite number of *breakpoints* at which one node of the network is going to change its status from active to inactive or vice versa. This concept is similar to the regularization path of Lasso where we can compute a finite number of breakpoints at which the active set changes.

To this end, we introduce a two-step iterative approach generally described as follows (see Fig. 2):

• **Step 1 (over-conditioning step).** Considering over-conditioning case by additionally conditioning on the selected pieces of all the hidden nodes in the DNN.

• **Step 2 (homotopy step).** Combining multiple over-conditioning cases by homotopy method to obtain $\mathcal{A}(\boldsymbol{x}(z))$ for all $z \in \mathbb{R}$.

### 3.1 STEP1: OVER-CONDITIONING STEP

We now show that by conditioning on the selected pieces $\boldsymbol{s}(\boldsymbol{x}^{\mathrm{obs}})$ of all the hidden nodes, we can write the selection event of the DNN as a set of linear inequalities.

**Lemma 1.** *Consider a class of DNN which consists of affine operations and piecewise-linear AFs. Then, the over-conditioning region is written as*

$$\mathcal{Z}^{\mathrm{oc}} = \{z \in \mathbb{R} \mid \Theta^{(\boldsymbol{s}(\boldsymbol{x}^{\mathrm{obs}}))}\boldsymbol{x}(z) \leq \boldsymbol{\psi}^{(\boldsymbol{s}(\boldsymbol{x}^{\mathrm{obs}}))}\}$$

*for a matrix $\Theta^{(\boldsymbol{s}(\boldsymbol{x}^{\mathrm{obs}}))}$ and a vector $\boldsymbol{\psi}^{(\boldsymbol{s}(\boldsymbol{x}^{\mathrm{obs}}))}$ which depend only on the selected pieces $\boldsymbol{s}(\boldsymbol{x}^{\mathrm{obs}})$.*

*Proof.* For the class of DNN, by fixing the selected pieces of all the piecewise-linear AFs, the input to each AF is represented by an affine function of an image $\boldsymbol{x}$. Therefore, the condition for selecting

a piece in a piecewise-linear AF, $s_j(\boldsymbol{x}(z)) = s_j(\boldsymbol{x}^{\mathrm{obs}})$, is written as a linear inequality w.r.t. $\boldsymbol{x}(z)$. Similarly, the value of each unit in the representation layer is also written as an affine function of $\boldsymbol{x}(z)$. Since the attention region is selected if the value is greater than a threshold, the choice of attention region $\mathcal{A}(\boldsymbol{x}(z)) = \mathcal{A}(\boldsymbol{x}^{\mathrm{obs}})$ is characterized by a set of linear inequalities w.r.t. $\boldsymbol{x}(z)$. □

Furthermore, let us consider max-operation, an operation to select the max one from a finite number of candidates. A max-operation is characterized by a set of comparison operators, i.e., inequalities. Let us consider a DNN which contains max-operators, and denote $\tilde{s}(\boldsymbol{x})$ be the set of selected candidates of all the max-operators for an input image $\boldsymbol{x}$.

**Corollary 1.** *Consider a class of DNN which consists of affine operations, max-operations and piecewise-linear AFs. Then, a region $\tilde{\mathcal{Z}}^{\mathrm{oc}}$ defined as $\tilde{\mathcal{Z}}^{\mathrm{oc}} := \{z \in \mathcal{Z}^{\mathrm{oc}} \mid \tilde{s}(\boldsymbol{x}(z)) = \tilde{s}(\boldsymbol{x}^{\mathrm{obs}})\}$ is characterized by a set of linear inequalities w.r.t. $\boldsymbol{x}(z)$.*

The proof is shown in Appendix A.1.

**Remark 1.** In this work, we mainly focus on the trained DNN where the activation functions used at hidden layers are *piecewise linear*, e.g., ReLU, Leaky ReLU, which is commonly used in CNN. Otherwise, if there is any specific demand to use non-piecewise linear functions such as sigmoid or tanh at hidden layers, we can apply some piecewise-linear approximation approach to these functions. We provided examples about the approximation for this case in Appendix A.5.

**Remark 2.** Most of the basic operations in a trained neural network are written as affine operations. In the traditional neural network, the multiplication results between the weight matrix and the output of the previous layer and its summation with bias vector is affine operation. In a CNN, the main convolution operation is obviously an affine operation. Upsampling operation is also affine.

**Remark 3.** Although the max-pooling operation is not an affine operation, it can be written as a set of linear inequalities. For instance, $v_1 = \max\{v_1, v_2, v_3\}$ can be written as a set $\{\boldsymbol{e}_1^\top \boldsymbol{v} \leq \boldsymbol{e}_2^\top \boldsymbol{v}, \boldsymbol{e}_1^\top \boldsymbol{v} \leq \boldsymbol{e}_3^\top \boldsymbol{v}\}$, where $\boldsymbol{v} = (v_1, v_2, v_3)^\top$ and $\boldsymbol{e}_i$ is a standard basis vector with a 1 at position $i$.

**Remark 4.** In Remark 1, we mentioned that we need to perform piecewise linear approximation for non-piecewise linear activations. However, if these functions are used at output layer, we *do not* need to perform the approximation task because we can define the set of linear inequalities based on the values before doing activation. See the next example for the case of sigmoid function.

**Example 1.** Let us consider a 3-layer neural network with $n$ input nodes, $h$ hidden nodes and $n$ ouput nodes. Let $W^{(1)} \in \mathbb{R}^{h \times n}$ and $\boldsymbol{w}^{(1)} \in \mathbb{R}^h$ respectively be the weight matrix and bias vector between input layer and hidden layer, and $W^{(2)} \in \mathbb{R}^{n \times h}$ and $\boldsymbol{w}^{(2)} \in \mathbb{R}^n$ respectively be the weight matrix and bias vector between hidden layer and output layer. The activation function at hidden layer is ReLU, and we use sigmoid function at output layer. At the hidden layer, for any node $j \in [h]$, the selection event is written as

$$
\begin{cases}
W_{j,:}^{(1)} \boldsymbol{x} + w_j^{(1)} \geq 0, & \text{if the output of ReLU function at } j^{\mathrm{th}} \text{ node} \geq 0, \\
W_{j,:}^{(1)} \boldsymbol{x} + w_j^{(1)} < 0, & \text{otherwise.}
\end{cases}
$$

Let $\boldsymbol{a}^{(1)} \in \mathbb{R}^h$ and $\boldsymbol{s}^{(1)} \in \mathbb{R}^h$ be the vectors in which $a_{j \in [h]}^{(1)} = 1, s_{j \in [h]}^{(1)} = 1$ if the output of ReLU function at the $j^{\mathrm{th}}$ node $\geq 0$, and $a_j^{(1)} = 0, s_j^{(1)} = -1$ otherwise. Then we have the linear inequality system $\Theta_1 \boldsymbol{x} \leq \boldsymbol{\psi}_1$ where $\Theta_1 = (-s_1^{(1)} W_{1,:}^{(1)}, ..., -s_h^{(1)} W_{h,:}^{(1)})^\top$ and $\boldsymbol{\psi}_1 = (s_1^{(1)} w_1^{(1)}, ..., s_h^{(1)} w_h^{(1)})^\top$. Next, for any output node $o \in [n]$, the selection event—a linear inequality—is written as

$$
\begin{cases}
W_{o,:}^{(2)}((W^{(1)} \boldsymbol{x} + \boldsymbol{w}^{(1)}) \circ \boldsymbol{a}^{(1)}) + w_o^{(2)} \geq 0, & \text{if the output of sigmoid function at } o^{\mathrm{th}} \text{ node} \geq 0.5, \\
W_{o,:}^{(2)}((W^{(1)} \boldsymbol{x} + \boldsymbol{w}^{(1)}) \circ \boldsymbol{a}^{(1)}) + w_o^{(2)} < 0, & \text{otherwise,}
\end{cases}
$$

where $\circ$ is the element-wise product. Similar to the hidden layer, we can also construct the linear inequality system $\Theta_2 \boldsymbol{x} \leq \boldsymbol{\psi}_2$ at the output layer. Finally, the whole linear inequality system is written as

$$
\Theta \boldsymbol{x} \leq \boldsymbol{\psi} = (\Theta_1 \, \Theta_2)^\top \boldsymbol{x} \leq (\boldsymbol{\psi}_1 \, \boldsymbol{\psi}_2)^\top. \tag{12}
$$

---

**Algorithm 1** `compute_solution_path`

---

**Input:** $\boldsymbol{a}, \boldsymbol{b}, [z_{\min}, z_{\max}]$
1: Initialization: $t = 1$, $z_t = z_{\min}$, $\mathcal{T} = z_t$
2: **while** $z_t < z_{\max}$ **do**
3:    Obtain $\mathcal{A}(\boldsymbol{x}(z_t))$ by applying a trained DNN to $\boldsymbol{x}(z_t) = \boldsymbol{a} + \boldsymbol{b} z_t$
4:    Compute the next breakpoint $z_{t+1} \leftarrow$ Equation (13). Then assign $\mathcal{T} = \mathcal{T} \cup \{z_{t+1}\}$, and $t = t + 1$
5: **end while**
**Output:** $\{\mathcal{A}(\boldsymbol{x}(z_t)\}_{z_t \in \mathcal{T}}$

---

### 3.2 STEP 2: HOMOTOPY STEP

We now introduce a homotopy method to compute $\mathcal{A}(\boldsymbol{x}(z))$ based on over-conditioning step.

**Lemma 2.** *Consider a real value $z_t$. By applying a trained DNN to $\boldsymbol{x}(z_t)$, we obtain a set of linear inequalities $\Theta^{(\boldsymbol{s}(\boldsymbol{x}(z_t)))} \boldsymbol{x}(z_t) \leq \boldsymbol{\psi}^{(\boldsymbol{s}(\boldsymbol{x}(z_t)))}$. Then, the next breakpoint $z_{t+1} > z_t$ at which the status of one node is going to be changed from active to inactive or vice versa, i.e., the sign of one linear inequality is going to be changed, is calculated by*

$$z_{t+1} = \min_{k:(\Theta^{(\boldsymbol{s}(\boldsymbol{x}(z_t)))} \boldsymbol{b})_k > 0} \frac{\psi_k^{(\boldsymbol{s}(\boldsymbol{x}(z_t)))} - (\Theta^{(\boldsymbol{s}(\boldsymbol{x}(z_t)))} \boldsymbol{a})_k}{(\Theta^{(\boldsymbol{s}(\boldsymbol{x}(z_t)))} \boldsymbol{b})_k}. \tag{13}$$

The proof is shown in Appendix A.2. Algorithm 1 shows our solution to efficiently identify $\mathcal{A}(\boldsymbol{x}(z))$. In this algorithm, multiple *breakpoints* $z_1 < z_2 < ... < z_{|\mathcal{T}|}$ are computed one by one. Each breakpoint $z_t, t \in [|\mathcal{T}|]$, indicates a point at which the sign of one linear inequality is changed, i.e., the status of one node in the network is going to change from active to inactive or vice versa. By identifying all these breakpoints $\{z_t\}_{t \in [|\mathcal{T}|]}$, the solution path is given by $\mathcal{A}(\boldsymbol{x}(z)) = \mathcal{A}(\boldsymbol{x}(z_t))$ if $z \in [z_t, z_{t+1}], t \in [|\mathcal{T}|]$. For the choice of $[z_{\min}, z_{\max}]$, see Appendix A.3.

## 4 EXPERIMENT

We highlight the main results. Several additional results and details can be found in Appendix A.6.

**Numerical Experiments.** We demonstrate the performances of two versions of the proposed method: *proposed-method* (homotopy) and *proposed-method-oc*. The $p$-values in these two versions were computed by (5) and (11), respectively. Besides, we also compared the proposed methods with the naive $p$-value in (4) and the permutation test. The details of permutation test procedure is described in Appendix A.6. To test the FPR control, we generated 120 null images $\boldsymbol{x} = (x_1, ..., x_n)$ in which $x_i \in [n] \sim \mathbb{N}(0, 1)$ for each $n \in \{64, 256, 1024, 4096\}$. To test the power, we generated images $\boldsymbol{x} = (x_1, ..., x_n)$ with $n = 256$ for each *true* average difference in the underlying model $\mu_{\mathcal{C}_{\boldsymbol{x}}^+} - \mu_{\mathcal{C}_{\boldsymbol{x}}^-} = \Delta_\mu \in \{0.5, 1.0, 1.5, 2.0\}$. For each case, we ran 120 trials. We chose the significance level $\alpha = 0.05$. For more information about the setup as well as the the structure of a neural network, see the experimental setup paragraph in Appendix A.6. The results of FPR control are shown in the first part of Fig. 3. The proposed methods could successfully control the FPR under $\alpha = 0.05$ while the naive method can *not*. Since the naive method fails to control FPR, we did not consider the power anymore. In the second part of Fig. 3, we see that the over-conditioning option has lower power than the homotopy method. It is because the truncation region in proposed-method-oc is shorter than the one in proposed-method (homotopy), which is demonstrated in the third part of Fig. 3. The last part of Fig. 3 shows the reason why the proposed homotopy method is efficient. With the homotopy method, we only need to consider the number of encountered intervals on the line along the direction of test statistic which is almost linearly increasing in practice.

**Real-data examples.** We performed comparison on real-world brain image dataset, which includes 939 images with tumor and 941 images without tumor. We first compared our method with permutation test in terms of FPR control. The results are shown in Table 1. Since the permutation test could not control the FPR properly, we did not compare the power. The comparisons between naive $p$-value and selective $p$-value are shown in Figs. 4, 5, 6 and 7. The naive $p$-value was still small

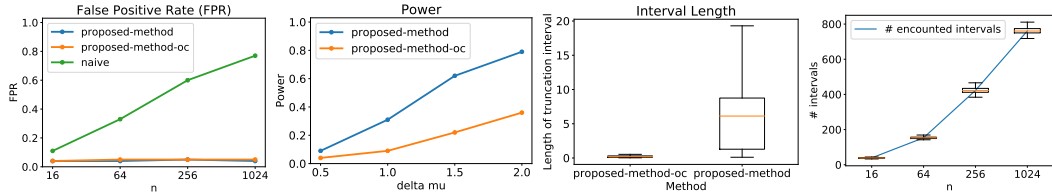

Figure 3: Results of false positive rate (FPR), power, length of interval, and encountered interval.

Table 1: FPR and power comparisons on real-world brain image dataset.

|  | Attention Detection Task | | Segmentation Task | |
| --- | --- | --- | --- | --- |
|  | FPR | Power | FPR | Power |
| Proposed Method | 0.056 | 0.669 | 0.057 | 0.683 |
| Permutation Test | 0.850 | – | 0.640 | – |

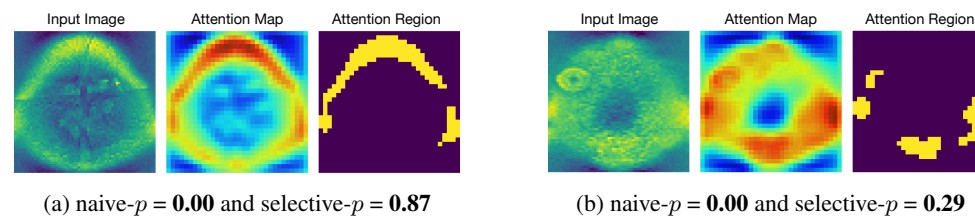

(a) naive-$p$ = **0.00** and selective-$p$ = **0.87**      (b) naive-$p$ = **0.00** and selective-$p$ = **0.29**

Figure 4: Inference on hypotheses obtained from internal representation (without tumor region).

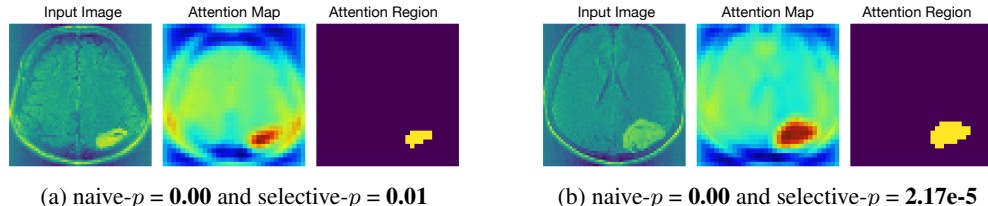

(a) naive-$p$ = **0.00** and selective-$p$ = **0.01**      (b) naive-$p$ = **0.00** and selective-$p$ = **2.17e-5**

Figure 5: Inference on hypotheses obtained from internal representation (with tumor region).

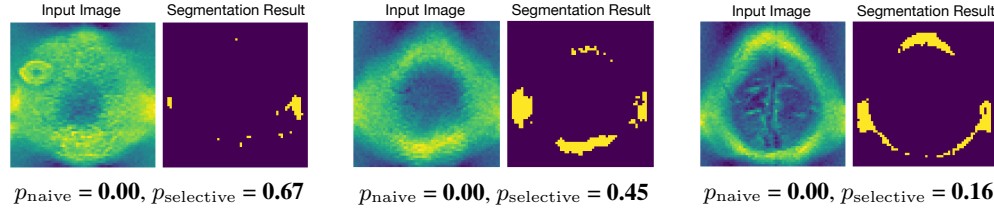

$p_{\text{naive}}$ = **0.00**, $p_{\text{selective}}$ = **0.67**      $p_{\text{naive}}$ = **0.00**, $p_{\text{selective}}$ = **0.45**      $p_{\text{naive}}$ = **0.00**, $p_{\text{selective}}$ = **0.16**

Figure 6: Inference on hypotheses obtained from output representation (without tumor region).

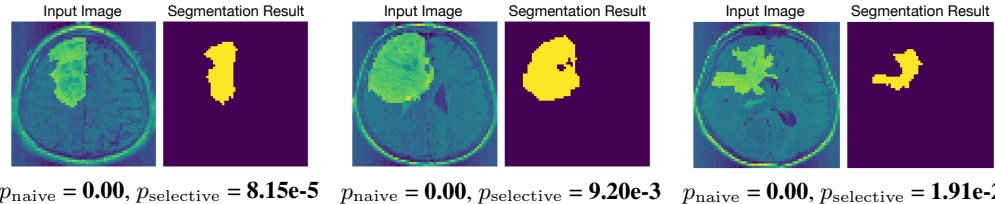

$p_{\text{naive}}$ = **0.00**, $p_{\text{selective}}$ = **8.15e-5**   $p_{\text{naive}}$ = **0.00**, $p_{\text{selective}}$ = **9.20e-3**   $p_{\text{naive}}$ = **0.00**, $p_{\text{selective}}$ = **1.91e-2**

Figure 7: Inference on hypotheses obtained from output representation (with tumor region).

even when the image has no tumor region, which indicates that the naive $p$-values cannot be used for quantifying the reliability of DNN-driven hypotheses. The proposed method could successfully identify false positive detections as well as true positive detections.

## 5 CONCLUSION

We proposed a novel method to conduct statistical inference on the significance of the data-driven hypotheses driven from neural network representation based on the concept of selective inference. In the context of explainable DNN or interpretable DNN, we are primarily interested in the reliability of the trained network when given new inputs (not training inputs). Therefore, the validity of our proposed method does *not* depend on how the DNN is trained.

In regard of the generality, the proposed method can be applied to any kind of network as long as the network operation is characterized by a set of linear inequalities (or approximated by piecewise-linear functions) because all the algorithms and theories in §2 and §3 only depend on the property of each component and does not depend on the entire structure of the network.

We believe that this paper provides a significant step toward reliable artificial intelligence (AI) and open several directions for *statistically* evaluating the reliability of DNN representation-driven hypotheses. Although it is not necessary to account the impact of training in this paper because the validity of our proposed method does not depend on how the DNN is trained, defining a new problem setup and providing solution for the case in which the training process needs to be considered is a potential direction. Moreover, widening the practical applicability of the proposed method in other fields such as NLP and signal processing would also represent a valuable contribution.

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
