# OpenReview forum: "Quantifying Statistical Significance of Neural Network Representation-Driven Hypotheses by Selective Inference"
_ICLR.cc/2021/Conference — Reject_

### Official Review · AnonReviewer4 · 2020-10-28
**Good work on reliability analysis of neural network representation learning**

**Rating:** 8
**Confidence:** 2

**Review:**

This paper proposed a novel method which to quantify the reliability of DNN-driven hypotheses in a statistical hypothesis testing framework. Naive statistical testings are not appropriate for the DNN-driven hypotheses, where the hypotheses are selected by looking at the data(i.e. The selection bias exists). To address this problem, the authors developed a novel homotopy method under the Selective-Inference(SI) framework, which can derive the exact sampling distribution of the DNN-driven hypotheses. In this paper,  the authors mainly focus on DNNs which consist of affine operations, max-operations, and piecewise-linear activation. As described by Lee et al. (2016), the main idea of SI is to make the inference conditional on the selection event. Specifically to the DNN-driven hypotheses, the authors proposed a novel method that consists of two steps, 1) Adding extra conditioning to make the problem traceable. 2) Combining multiple over-conditioning cases by homotopy method to solve the over-conditioning problem. The experimental results on both synthetic and real-world datasets illustrate the proposed method can successfully control the FP error rate.

Considering that there are more and more interests in the research on neural representation learning, the problem that this work is trying to solve is pretty important. The SI framework has been well studied on the lasso and other problems. To the best of my knowledge, this is the first work that deploys SI to test DNN representation driven hypotheses. Although the authors only demonstrate it on simple DNNs, I can see the potentials of this method to apply on more practical and complex DNNs. The community may benefit from it on understanding neural representations.

Pros:
- The problem that this paper is trying to solve, is clearly defined and is essential in understanding the representations DNN learned.
- This proposed SI algorithm based on the homotopy method can derive the exact conditional sampling distribution of DNN-driven hypotheses in an efficient way, and it is proved to be effective in practical by experimental results. It also shows its novelty in solving the problem.
- The authors provided comprehensive supplementary materials to help readers to understand the proposed method as well as to reproduce the experiments.
- The structure of the paper is well designed, and the writing is clear.

Cons:
- While this paper claims that the proposed method can quantify the reliability of neural network representation-driven hypotheses, there could be more examples (more realistic models on more tasks) to demonstrate the method's effectiveness in more scenarios. Currently, the examples in this paper are almost all using basic NN components on image inputs. The performances are not obvious if the networks contain parts like residual connections or recurrent structures, either the inputs are sentences rather than images.
- Related to the previous point, the boundary between where this method can be applied and can not be applied is not so clear. It would be better if the authors could give such guidance for people to use this method.

---

> ### Author Response · Authors · 2020-11-15
> **Our responses to Reviewer 4**
>
> Thank you for your detailed review and positive assessment.
>
> We are encouraged by hearing your opinion that the problem that we are trying to solve is very important for DNN community.
>
> We have revised the paper to clarify the points you raised in your review. Here are the answers to your questions.
>
> 1. The generality of the proposed method (recurrent structures, inputs are sentences rather than images, etc.)
>
> We added a discussion about this point in second paragraph of Section 5 in the revised paper.
>
> Although we used computer vision tasks as concrete examples, all the algorithms and theories in sections 2 and 3 are generally applicable to other tasks.
>
> Furthermore, since the applicability of the proposed method does not depend on the entire structure of the network but on the property of each component, it can be applied as long as the each component of the network can be characterized by a set of linear inequalities (or approximated by piecewise-linear functions). Also for recurrent structure, as long as we can decompose the operations into a set of piece-linear function operations, it should be fine.
>
> 2. Where this method can be applied and can not be applied.
>
> The main conditions for the applicability of the proposed method are
>
> a) the operations can be written as a set of linear inequalities or can be accurately approximated by piecewise linear approximation.
>
> b) the test-statistic is written as a linear contrast of input vector
>
> We observe that many commonly-used practical DNN structures satisfy the first condition a).
>
> A nice thing about our "extra conditioning + combining" approach is that, even the network is large and complex, we can decompose it into a set of small components such that each component can be accurately approximated by a linear operation.
>
> Although we only provided examples in computer vision tasks in this paper, we believe that the problem setup is fairly general. We guess that there are various practical problems that fit into our approach in other fields such as NLP, signal processing, and control.
>
>
> We are grateful for your deep understanding of the main message of our paper.
>
> If you have any further questions, we would be happy to discuss with you during the discussion period.

---

### Official Review · AnonReviewer3 · 2020-10-28
**Important problem and nice approach**

**Rating:** 7
**Confidence:** 2

**Review:**

The authors present a new method to quantify the representational strength of the neural networks. In particular, they address a known issue about how to judge a network's result relative to a hypothesis if the source of both the hypothesis and the network training was the same dataset.
I believe the goal and motivation of this paper are clear and very well useful. Having a robust and fair theoretical framework would be a good step in analyzing a lot of vision approaches and approaches in NLP, control, and others. They also give some promising experimental results.
The techniques used in this work (e.g., selection of extra conditioning for traceability, Merging results of condition sets) have a satisfactory amount of novelty and can be of independent interest to the ML community. Since their approach (at least partially) addresses the issue explained above, it may find more interest among researchers of downstream tasks for explainability and reliability evaluation.
I suggest the authors address famous problems with frequentist statistics and commonly misinterpreted p-value. I want to see the author's opinion about the position of their approach relative to such issues. Perhaps having a discussion section or limitations of the study would help. Also, does this approach have a Bayesian counterpart so that it doesn't rely on p-values?

---

> ### Author Response · Authors · 2020-11-15
> **Our responses to Reviewer 3**
>
> Thank you for your positive and encouraging assessment of our work.
>
> It is great to hear your opinion that the target problem is important and our approach is interesting.
>
> In the revised paper, following your comments, we added a discussion on the assessment of explainability and reliability using p-values.
>
> 1. Opinion on misinterpreted p-values.
>
> In the revised paper, the discussion was added in latter part of Paragraph 3 in Section 1.
>
> We agree that p-values are often misinterpreted. There are various source of mis-interpretation as discussed in the following statement from ASA (American Statistical Association).
>
>   ASA (American Statistical Association) statement on p-values. https://www.amstat.org/asa/files/pdfs/P-ValueStatement.pdf
>
> In this paper, by using selective inference, we address one of the sources of mis-interpreted p-values; the p-values are biased when the hypothesis is selected after looking at the data (often called double-dipping or data dredging). We believe that our approach is a significant first step to find "valid p-values" in assessing the reliability of DNN representations.
>
>
> 2. Does the proposed approach have a Bayesian counterpart so that it doesn't rely on p-values?
>
> There are a few studies trying to correct the selection bias in Bayesian model selection based on the similar concept as selective inference (e.g., see section 6.3 of Bi et al.(2017)).
>
>   Bi et al., Inferactive data analysis. arXiv:1707.06692
>
> However, we have not pursued this direction yet and we are not sure such an approach can be applied, e.g., to Bayesian DNN. To the best of our knowledge, there has been no such discussion in the literature.
>
>
> We appreciate your high-level discussion regarding frequentist vs. Bayesian approaches for explainability and reliability evaluation of DNN.
>
> We welcome further discussion with you during the discussion period.

---

### Official Review · AnonReviewer2 · 2020-10-29
**A nice application of selective inference to finding statistically significant attention area of an image**

**Rating:** 6
**Confidence:** 3

**Review:**

The authors propose a framework of selective inference to rigorously study the problem of finding significant attention areas of a neural network model (with certain constraints).

Pros
- The proposed method yields good performance in power in practice.

- The FDR of the proposed method is theoretically controlled.

- The authors proposed an efficient algorithm to find certain kinds of attention areas.

Cons & Questions:

- Is the assumption (1) necessary for the proposed non-asymptotic method? Please address. If it is necessary, please also address why this assumption is practical, or existing work where the same was assumed.

- The proposed method has several assumptions on the neural network, such as the layers need to be affine / max pooling / piecewise linear. Given the design of popular neural networks, the method is widely applicable enough. However, it will be helpful if the authors can address the commonly used network structure where the proposed method cannot be used.

For example, can one uses this method to test the alternative hypothesis that a region found by BERT's internal attention mechanism is significant? Such a case study will help readers gain more clarity of the method.

Also, if the method is not applicable to softmax layer, does the score produced from the logit? Please clarify.

---

> ### Author Response · Authors · 2020-11-15
> **Our responses to Reviewer 2**
>
> Thank you for your positive assessment of our work.
>
> We have revised the paper to clarify the points you raised in your review. Here are the answers to your questions.
>
> 1. Is the assumption (1) necessary for the proposed non-asymptotic method?
>
> In Equation (1), we assumed the noise vector added to pixels follows a multivariate Normal distribution. In computer vision and other signal processing problems, it is not uncommon to assume Gaussian noise. This corresponds to assuming Normally-distributed noise in hypothesis testing for linear regression model, which we believe is reasonable and common.
>
> Note that hypothesis testing requires a certain probability model for explaining random variation. All existing selective inference studies such as Lee+ (2016), Fithian+ (2014), Tibshirani+ (2016) and Liu+ (2018) also use the same kind of assumptions.
>
> It is, of course, important to check whether this assumption is appropriate or not for each application task. Furthermore, the method is desired to be robust against violations from the assumption. To check the robustness of the method; see Appendix A.6 for robustness analysis of the proposed method.
>
> 2. Commonly used network structure where the proposed method cannot be used.
>
> Our proposed method can be applied to any kind of network as long as the network operation is characterized by a set of linear inequalities. We added this discussion in Section 5 of the revised paper. Furthermore, as illustrated in Appendix A.5, by approximating nonlinear operations with piecewise linear functions, we have confirmed that statistical inference can be made with sufficient accuracy even in networks which contains nonlinear operators.
>
> Since the applicability of the proposed method does not depend on the entire network structure, but on the property of each component, the proposed method should be applicable to large scale networks.
>
> One practical issue we are facing right now is the high implementation cost. To implement the proposed SI framework for a large and complex network such as Bert, we have to implement all the events for all the components in the network from scratch. Our future plan is first to implement the proposed method for a few commonly used network structures, and then to develop a generic software tool that can automatically compute the selection event for selective inference when the network structure is given.
>
> 3. Applicability to softmax layer
>
> As we demonstrated in Appendix A.5, if we can approximate nonlinear operations by piecewise-linear functions with sufficiently large number of pieces, the method can cover the nonlinear operations. We have not tested whether this approach practically works well in the case of softmax layer.
>
>
> Thank you again for your encouraging comments in your review.
> We are happy to discuss if you have further questions.

---

### Official Review · AnonReviewer1 · 2020-10-30
**Paper1095 review**

**Rating:** 6
**Confidence:** 4

**Review:**

### Summary

In this paper, the authors propose a novel approach for quantifying the statistical significance of binary masks predicted by a subclass of deep neural network (DNN) models for image segmentation problems.

In brief, the manuscript considers the particular setting where a model has been pretrained to produce a binary output of the same dimension as the input, which can be interpreted as a binary attention mask / image segmentation output. In this scenario, the main contribution is an approach to test the null hypothesis that a linear contrast of the image, depending on the segmentation outputs, equals zero. In particular, the specific formulation put forward in the paper tests the null hypothesis that the average value of pixels classified as 1 (e.g. in the mask) is equal to the average value of pixels classified as 0 (e.g. not in the mask).

Since the linear contrast is a function of the input, in this case, a nonlinear function implemented by the pre-trained DNN model, classical statistical inference (such as e.g. a two-sided t-test) does not apply. Instead, the authors propose to leverage recent advances in the field of selective inference to derive the null distribution of the test statistic conditioned on the DNN’s output (plus additional constraints to get rid of nuisance parameters).

In a nutshell, to accomplish this, they:

1) Assume that images follow a multivariate normal distribution with known covariance matrix,
and
2) Restrict the class of DNN models to those whose internal representation in every layer can be expressed as a piecewise affine function of the input. Namely, this excludes activation functions other than piecewise linear functions (e.g. ReLU), which the authors propose to approximate using piecewise linear functions.

The modelling assumption (1) together with the model constraints (2) permit the problem to be cast in a form for which the celebrated Polyhedral lemma (Lee et al. 2016) applies.

Finally, the authors also tackle the algorithmic problem of efficiently integrating over the resulting truncation regions. To this end, they propose two approaches: one based on over-conditioning, inspired by Lee et al. 2016, and another based on the homotopy method, possibly inspired by Liu et al. 2018.

Experimental results are provided for a synthetic toy problem with “images” that follow a multivariate normal distribution with no correlation between pixels. The ground-truth attention mask is a large square region in the center of the image, and pixels within this region have a larger mean than those outside. DNN models consisting of 2 convolutional layers are pretrained on data generated under the same distribution, with the entire ground-truth binary segmentation masks used for training for each “image”. Results suggest that, under these conditions, the proposed approach succeeds in controlling the type I error while achieving non-trivial statistical power. The supplementary material shows that type I error remains controlled for other distributions such as Laplace or Student-t with 20 DoF.

Finally, results are also shown for a real-world brain image dataset, where type I error appears to be only sensibly larger than the tower and power non-trivial as well.

### High-level assessment

From a methodological perspective, the authors have identified a novel, interesting application of existing work, namely, the now seminal work of Lee et al. 2016 and recent improvements over those ideas presented by Liu et al. 2018, to the problem of image segmentation using certain DNN models. Moreover, they also proposed a sound algorithmic implementation of these ideas targeting the application at hand and provided some nascent, preliminary results that suggest the proposed approach might be of use in certain applications such as medical imaging.

All in all, this represents an original contribution of relevance to the field that I wish to see published eventually. However, I believe the current version of the manuscript falls short in certain areas, and should be improved prior to publication. Mainly:

1) I have strong reservations regarding the adequacy of the modelling assumptions required by the selective inference framework for this application.

2) The experimental results are substantially lacking, both in terms of breadth and depth.

3) The paper is also lacking in clarity. Many key, low-level aspects such as model architecture, (pre)-training procedure or assumptions between the relation of pre-training and testing data are either not clearly stated or mentioned without proper justification or discussion of its implications and impact on the overall results. Likewise, I feel that some of the limitations of the proposed approach are understated and that its generality is overstated, especially in the abstract and the introduction.

Because of this, I am as of now leaning towards recommending rejection, as I believe the paper would greatly benefit from a strong, non-incremental revision.

### Major points / suggestions for improvement

1) In order to frame the problem within the assumptions of the Polyhedral lemma, the authors postulate that images in the dataset follow a multivariate normal distribution, with unknown mean but known covariance matrix.

Unfortunately, while selective inference is known to abound on strong parametric assumptions, I believe this assumption is just too strong and hard to justify in the context of image data relative to, say, interpreting the coefficients of a linear model as done in more “typical” applications of selective inference.

While it might be unrealistic to expect the authors to overcome inherent limitations of the current state of the art in selective inference, I believe they should at least carry out a much more thorough investigation of the extent to which these assumptions are applicable to real-world data, and what are the consequences of violations for the inference process.

The robustness results provided in the supplementary material are a good step in this direction, but keep many unrealistic assumptions, such as unimodality or simplistic / inexistent correlation structures that remain far from representative of natural or medical images.

2) The proposed approach relies on pre-training the DNN model on a separate dataset, since the authors have made no attempt to account for the impact of training itself on the null distribution.

This can be an important limitation, especially in applications where data does not abound, which unfortunately might have a strong overlap with applications for which statistical significance might be of special relevance. Moreover, the need to split the data into “development” and “inference” sets will introduce randomness of the inference results, apart from incurring a loss in statistical power.

To this end, I believe the authors should: (1) discuss in much greater clarity how the pre-training of the DNN models interacts with their proposed approach; (2) study the extent to which the resulting inferences are robust to data splitting & retraining of the DNN model and (3) the impact that data splitting has on the resulting statistical power.

3) The paper is substantially lacking in clarity, leaving the reader to second-guess many key aspects of the proposed approach.

One such example is precisely the issue of pre-training. It is never clearly stated in the main manuscript how the DNN models are (pre)-trained, and what the relation between the training and inferences datasets is exactly.

Other, related aspects are (i) which type of supervision the models require for pre-training and (ii) which type of output layers are required in order for the null hypothesis to be relevant. For example, I had to look at the code in order to know that, in their simulation experiments, models are trained with the segmentation masks as supervision targets. To the best of my knowledge, this is nowhere stated in the paper. Similarly, I am uncertain of how the models for the real-world brain image dataset were trained. Given the change in architecture reported in the supplementary material, I could guess that in this case only tumor/no tumor labels were provided, but this is neither stated nor, as far as I could see, shown in the code provided. Also, in the latter case, I imagine that in order to keep the null hypothesis relevant, the model must rely on a global average pooling layer prior to classification since, otherwise, a null hypothesis defined in terms of the difference of pixel intensity values would not capture well the behavior of the output layer(s). More generally, the choice of architecture for the models is not justified in any way.

As a final example, it is also never precisely stated how exactly the output representation is thresholded in order to define the binary attention mask, how the threshold parameter is supposed to be chosen by a user and what its impact is in overall performance.

All in all, in my opinion, at the moment the manuscript does not provide sufficient information for a reader to accurately reproduce its results from the text alone.

4) In my opinion, the abstract and introduction currently overstate the generality of the proposed approach. I believe that claims such as “testing the reliability of DNN representations” implicitly imply the approach can be applied to any kind of DNN model/problem. However, the target application for the framework here developed is narrower and applies only to image segmentation problems and perhaps to other strongly related settings (e.g. problems where the target representation has the same size as the input and can be interpreted as an attention mask of sorts).

To clarify, I do not intend to imply the author’s contribution is insufficient. I believe proposing an approach to quantify the statistical significance of image segmentation masks is on its own a worthwhile contribution. However, I do believe it would be preferable that the abstract and introduction were written in a way that was more specific to what the manuscript truly proposes, implements and tests.

5) The experimental results, while providing some encouraging preliminary results supporting the proposed approach, in my opinion fall short in multiple aspects.

Firstly, as mentioned in point 1. above, I believe the experiments on synthetic data as they stand now are barely a “sanity-check” for the model. Instead, I suggest that the authors explore ways to create more challenging synthetic datasets capturing characteristics of real-world data, perhaps (but not necessarily) by exploiting deep generative models trained on natural / imaging data. In brief, the goal should be to exhaustively characterize (1) the extent to which the modelling assumptions required by selective inference are applicable to real-world data and (2) which failure modes / limitations the proposed approach might have.

Secondly, the models used by the authors are too small and not representative of those deployed in the target applications. I would encourage the authors to show results pertaining larger models (e.g. ResNet-based architectures with > 10 layers).

Finally, related to the previous point, it would be of interest to explore the effect of model size & architecture (depth, number of units per layer, presence of residual connections / normalization) on the number of intervals encountered, false positive rate and statistical power.

### Minor points (not related to the manuscript’s rating)

1) In my opinion, Example 1 would be much more informative if it included inactive units rather than the special, much easier case where all hidden units are active.

2) Personally, I do not find the definition of power in the supplementary material to be sufficiently clear. I would encourage the authors to define mathematically what # detected and # rejected are in their context.

3) To the best of my knowledge, the code does not currently set the seed for the PRNG, which might be detrimental for reproducibility.

4) I would recommend having the submission proof-read for English style and grammar issues.

---

> ### Author Response · Authors · 2020-11-15
> **Our responses to Reviewer 1**
>
> Thank you for your detailed and constructive review. Your feedback will greatly help us to improve our paper.
>
> We updated the paper according to your comment and will keep polishing the paper during the rebuttal period.
>
> In the following, let us respond to your five major points.
>
> 1. The assumption that an image follows a multivariate normal distribution.
>
> We would like to emphasize that we do NOT assume that pixel intensities in an image follow Normal distribution. In Equation (1), we just assume that the vector of noises added to the true pixel values follows a multivariate Normal distribution. In computer vision and other signal processing problems, it is not uncommon to assume Gaussian noise. We added the text description in the first paragraph of Section 2.
>
> That being said, we totally agree that the robustness against violation from the Normality assumption is desirable. That is the reason why, in Appendix A.6, we provided robustness analyses of the proposed method in terms of the FPR control when data follows skew normal distribution, Laplace distribution, t-distribution and sigma is estimated from the data. However, we would highly appreciate if the reviewer understand that robustness analysis is not the main target in our paper.
>
> 2. The proposed approach relies on pre-training the DNN model on a separate dataset.
>
> Our goal is to provide the reliability of “post-hoc attentions” such as CAM (Zhou+., 2016) or Grad-CAM, in which the method is applied to a DNN after the training is completed and the parameters are fixed. Recent papers discussing the reliability of the representations that are cited in the first paragraph of Section 1 point out that there are issues in the stability and reproducibility of post hoc attentions.
>
> We agree that providing reliability of the representation for a training input is more challenging because the selection bias stemming from the training mechanism must be properly incorporated. However, to our knowledge, in the context of “explainable AI” research, we are primarily interested in the reliability of the trained network when given new inputs (not training inputs).
>
> We did not discuss the details of the pre-trained DNNs and data-splitting because the validity of our proposed method does NOT depend on how the DNN is trained, as long as the target input for the inference is independent of the training set. In the revision, we clarified this point.
>
> 3. Lacking in clarity
>
> In regard to the text description of the pre-training process, we added the information that you suggested in experimental setup part of Appendix A.6. For the details of how the model for brain image dataset is trained, please refer to our code (train_tumor_data.py). Regarding the relation between the training and inferences, as we discussed above, the validity of our method does not depend on the training process.
>
> Regarding the minor points, we also updated the Example 1 by including inactive units. Besides, we also provided the descriptions for # detected and # rejected in the definition of the power Appendix A.6.
>
> 4. The generality of the proposed method
>
> We added a discussion in second paragraph of Section 5 in the revised paper. Although we used computer vision tasks as concrete examples, all the algorithms and analysis in sections 2 and 3 can be applied to other problems as long as the data has the form in (1), the DNN operation can be represented as stated in Lemma 2 and 3, and the test-statistic is written as a linear contrast of the input vector.
>
> Following your suggestion, in the revised abstract, we clearly mentioned that “In this paper, we demonstrate the proposed method in computer vision tasks as practical examples”.
>
> 5. Extension to more challenging synthetic datasets and larger DNN models
>
> We completely agree with your opinion that it is worth working on large-scale problems. Since the applicability of our algorithm and analyses depends not on the entire network structure but on the components of the network, it should be able to be applied to large-scale network as long as the components satisfy the conditions.
>
> We believe that the experiments on brain image classification and segmentation are simple yet practical demonstrations since the DNN used in the experiments actually have good enough classification and segmentation performances on the real brain image datasets. Even with this simple network, to our knowledge, it was not possible to provide the reliability of the representation in the form of p-values.
>
> Thank you again for reading our paper very carefully. We hope that our replies  address your concerns and questions.
>
> We believe that our work is a significant step toward reliable AI in that we are the first to introduce selective inference to provide the reliability of DNN representation in the form of exact p-values, although we understand that there remain some more challenges for generalizations and extensions.
>
> We welcome further discussion during the discussion period.

---

### Decision · Program_Chairs · 2021-01-07
**Final Decision**

**Decision:**

Reject

**Comment:**

Four reviewers evaluated your work and provided a detailed review with many suggestions. I also think that there is an interesting idea and encouraging results but there is a lack of numerical results and still some parts are still unclear and need to be polished. Consequently in its  current form, the paper can not be accepted for publication. I would advise you to carefully follow the remarks of reviewer 1 to improve  the paper.